

# OBSERVATIONS OF HIGHLY OXIDISED MOLECULES AND PARTICLE NUCLEATION IN THE ATMOSPHERE OF BEIJING

**James Brean[1], Roy M. Harrison[1*†], Zongbo Shi[1] David C.S. Beddows[1], W. Joe F. Acton[2] and C. Nicholas Hewitt[2]**

**[1]Division of Environmental Health and Risk Management, School of Geography, Earth and Environmental Sciences University of Birmingham Edgbaston, Birmingham B15 2TT United Kingdom**

**[2]Lancaster Environment Centre Lancaster University, Lancaster LA1 4YQ United Kingdom**

---

* To whom correspondence should be addressed.
Tele: +44 121 414 3494; Fax: +44 121 414 3709; Email: r.m.harrison@bham.ac.uk

†Also at: Department of Environmental Sciences / Center of Excellence in Environmental Studies, King Abdulaziz University, PO Box 80203, Jeddah, 21589, Saudi Arabia



**ABSTRACT**
Particle nucleation is one of the main sources of atmospheric particulate matter by number, with new
particles having great relevance for human health and climate. Highly oxidised multifunctional
organic molecules (HOMs) have been recently identified as key constituents in the growth, and,
sometimes, in initial formation of new particles. While there have been many studies of HOMs in
atmospheric chambers, flow tubes and clean environments, analyses of data from polluted
environments are scarce. Here, measurements of HOMs and particle size distributions down to small
molecular clusters are presented alongside VOC and trace gas data from a campaign in Beijing. Many
gas phase HOMs have been characterised and their temporal trends and behaviours analysed in the
context of new particle formation. The HOMs identified have a comparable degree of oxidation to
those seen in other, cleaner, environments, likely due to an interplay between the higher temperatures
facilitating rapid hydrogen abstractions and the higher concentrations of NOx and other $RO_2^.$
terminators ending the autoxidation sequence more rapidly. Our data indicate that alkylbenzenes,
monoterpenes, and isoprene are important precursor VOCs for HOMs in Beijing. Many of the $C_5$ and
$C_{10}$ compounds derived from isoprene and monoterpenes have a slightly greater degree of average
oxidation state of carbon compared to those from other precursors. Most HOMs except for large
dimers have daytime peak concentrations, indicating the importance of $OH^.$ chemistry in the
formation of HOMs, as $O_3$ is lower on the days with higher HOM concentrations; similarly, VOC
concentrations are lower on the days with higher HOM concentrations. The daytime peaks of HOMs
coincide with the growth of freshly formed new particles, and their initial formation coincides with
the peak in sulphuric acid vapours, suggesting that the nucleation process is sulphuric acid-dependent,
with HOMs contributing to subsequent particle growth.




## 1.    INTRODUCTION


Atmospheric particle nucleation, or the formation of solid or liquid particles from vapour phase
precursors is one of the dominant sources of global aerosol by number, with primary emissions
typically dominating the mass loadings (Tomasi et al., 2016). New particle formation (NPF) or the
secondary formation of fresh particles is a two-step process comprising of initial homogeneous
nucleation of thermodynamically stable clusters and their subsequent growth. The rate of growth
needs be fast enough to out-compete the loss of these particles by coagulation and condensation
processes in order for the new particles to grow, and hence NPF is a function of the competition
between source and sink (Gong et al., 2010). New particle formation has been shown to occur
across a wide range of environments (Kulmala et al., 2005). The high particle load in urban
environments was thought to suppress new particle formation until measurements in the early 2000s
(McMurry et al., 2000; Shi et al., 2001; Alam et al., 2003), with frequent occurrences observed even
in the most polluted urban centres. NPF events in Beijing occur on about 40% of days annually,
with the highest rates in the spring (Wu et al., 2007, 2008; Wang et al., 2016).  Chu et al. (2019)
review the many studies of NPF which have taken place in China and highlight the need for long-
term observations and mechanistic studies.

NPF can lead to production of cloud condensation nuclei (CCN) (Wiedensohler et al., 2009; Yu and
Luo, 2009; Yue et al., 2011; Kerminen et al., 2012) which influences the radiative atmospheric
forcing (Penner et al., 2011). A high particle count, such as that caused by nucleation events, has
been shown to precede haze events in environments such as Beijing (Guo et al., 2014). These events
are detrimental to health and quality of life. The sub-100 nm fraction of particles to which new
particle formation contributes to is often referred to as the ultrafine fraction. Ultrafine particles
(UFPs) pose risks to human health due to their high number concentration. UFPs exhibit gas-like
behaviour and enter all parts of the lung before penetrating into the bloodstream (Miller et al.,
2017). They can initiate inflammation via oxidative stress responses, progressing conditions such as



atherosclerosis and initiating cardiovascular responses such as hypertension through to myocardial
infarction (Delfino et al., 2005; Brook et al., 2010).

Highly oxidised multifunctional molecules (HOMs), organic molecules with O:C ratios >0.6, are
the result of atmospheric autoxidation and have recently been subject to much investigation, in part
because the extremely low volatilities arising from their high O:C ratios favour their condensation
into the particulate phase. HOMs are most well characterised as the product of oxidation of the
biogenic monoterpenoid compound $\alpha$-pinene (Riccobono et al., 2014; Tröstl et al., 2016; Bianchi et
al., 2017). Although globally, BVOC concentrations far exceed aromatic VOC concentrations by
approximately a factor of 10, in the urban environment the aromatic fraction is far more significant.
Formation of HOMs from aromatic compounds has been demonstrated in laboratory studies and
these have been hypothesised to be large drivers of NPF in urban environments (Wang et al., 2017;
Molteni et al., 2018; Qi et al., 2018). The formation of HOMs through autoxidation processes
begins with the reaction of VOCs with $OH^.$, $O_3$ or $NO^._3$; formation of a peroxy radical ($RO^._2$) is
followed by rapid $O_2$ additions and intra-molecular hydrogen abstractions (Jokinen et al., 2014;
Rissanen et al., 2014; Kurtén et al., 2015). Furthermore, generation of oligomers from stabilised
Criegee intermediates arising from short chain alkenes has been hypothesised as a contributor of
Extremely Low Volatility Organic Compounds (ELVOCs) and Low Volatility Organic Compounds
(LVOCs) (Zhao et al., 2015). The result of the large size and numerous oxygen-containing
functionalities in all of these compounds is a low vapour pressure, and therefore they make a
significant contribution to particle growth (Tröstl et al., 2016), although the contribution of HOMs
to the initial molecular clusters is still debated (Kurtén et al., 2016; Elm et al., 2017; Myllys et al.,

92   2017).


Recent technological advances have facilitated insights into the very first steps of nucleation which
were previously unseen, with mass spectrometric techniques such as the Atmospheric Pressure



Interface Time of Flight Mass Spectrometer (APi-ToF) and its chemical ionisation counterpart (CI-APi-ToF) allowing for high mass and time resolution measurements of low volatility compounds and molecular clusters. Diethylene glycol based particle counters, such as the Particle Size Magnifier (PSM) allow for measurements of particle size distributions down to the smallest molecular clusters nearing 1 nm. Recent chamber studies have elucidated the contribution of individual species to particle nucleation, ammonia and amines greatly enhancing the rate of sulfuric acid nucleation (Kirkby et al., 2011; Almeida et al., 2013). In these studies, HOMs have been identified, formed through autoxidation mechanisms (Schobesberger et al., 2013; Riccobono et al., 2014; Ehn et al., 2014). These are key to early particle growth (Tröstl et al., 2016) and can nucleate even in the absence of sulfuric acid in chambers (Kirkby et al., 2016) and in the free troposphere (Rose et al., 2018). In this paper, we report the results of HOM and particle size measurements during a summer campaign in Beijing, China.

## 2.    DATA AND METHODS

### 2.1.    Sampling Site

Sampling was performed as part of the APHH-Beijing campaign, a large international collaborative project examining emissions, processes and health effects of air pollution. For a comprehensive overview of the programme, see Shi et al. (2018). All sampling was conducted across a one month period at the Institute for Atmospheric Physics (IAP), Chinese Academy of Sciences, Beijing (39°58.53'N, 116°22.69'E). The sampling was conducted from a shipping container, with sampling inlets 1-2 metres above ground level, the nearest road being 30 metres away. Meteorological parameters (wind speed, wind direction, relative humidity (RH) and temperature) were measured at the IAP meteorological tower, 20 metres away from the sampling site, 30 metres from the nearest road at a height of 120 metres. Data was continuously taken from the CI-APi-ToF during a two week period, but due to data losses only five days of data is presented here. Particle size distribution measurements were taken during a 33 day period from 24/05/2017 – 26/06/2017.



## 2.2 Chemical Ionisation Atmospheric Pressure Interface Time of Flight Mass

### Spectrometry

The Aerodyne Nitrate Chemical Ionisation Atmospheric Pressure Interface Time of Flight Mass

Spectrometer (CI-APi-ToF) was used to make measurements of neutral oxidised organic

compounds, sulfuric acid and their molecular clusters at high time resolution with high resolving

power. The ionization system charges molecules either by forming an adduct with $NO_3^-$, or by

proton transfer to $NO_3^-$. The former occurs largely with species with two hydrogen bond donor

groups, such as organics with two or more hydroxyl or hydroperoxyl functionalities (Hyttinen et al.,

2015), with hydroperoxyl being the more efficient hydrogen bond donor (Møller et al., 2017).

Proton transfer occurs with molecules with great proton affinity such as sulfuric acid, although

clustering with sulfuric acid does occur. This instrument has been explained in great detail

elsewhere (Junninen et al., 2010; Jokinen et al., 2012), but briefly the front end consists of a

chemical ionisation system where a 10 Lpm sample flow is drawn in through the 1 metre length 1"

OD stainless steel tubing opening. A 3 sccm flow of a carrier gas ($N_2$) containing nitrate ions

generated by the X-ray ionisation of nitric acid vapour is run parallel and concentric to the sample

flow in an ion reaction tube. The nitrate ions will then charge molecules either by clustering or

proton transfer. The mixed flows travelling at 10 sLm enter the critical orifice at the front end of the

instrument at 0.8 sLm and are guided through a series of differentially pumped chambers before

reaching the ToF analyser. Two of these chambers contain quadrupoles which can be used to select

greater sensitivity for certain mass ranges, and the voltages across each individual chamber can be

tuned to maximise sensitivity and resolution for ions of interest. Mass spectra are taken at a

frequency of 20 kHz but are recorded at a rate of 1 Hz. All data analysis was carried out in the

*Tofware* package in *Igor Pro 6* (Tofwerk AG, Switzerland). A seven point mass calibration was

performed for every minute of data, and all data was normalised to signal at 62, 80 and 125 *m/Q* to

account for fluctuations in ion signal, these masses representing $NO_3^-$, $H_2ONO_3^-$ and $HNO_3NO_3^-$

respectively. The nitrate-water cluster is included as the presence of many nitrate-water clusters of



the general formula $(H_2O)_x(HNO_3)_yNO_3^-$ were found, where $x = (1, 2, 3... 20)$ and $y = (0, 1)$. No
sensitivity calibration was performed for these measurements, and so all values are reported in
signal intensity, ions/s. Due to the high resolving power of the CI-APi-ToF system, multiple peaks
can be fit at the same unit mass and their molecular formulae assigned. These peaks follow the
general formula $C_xH_yO_zN_w$ where $x = 2-20$, $y = 2-32$, $z = 4-16$ and $w = 0-2$, spanning from small
organic acids like oxalic and malonic acid through to large dimers of oxidised monoterpene $RO_2^{\cdot}$
radicals such as $C_{20}H_{31}O_9N$. Beyond 500 $m/Q$, peak fitting and assignment of compositions
becomes problematic as signal decreases, mass accuracy decreases, and the total number of
chemical compositions increases, so peaks above the $C_{20}$ region have not been assigned, and a
number of peaks have been unassigned due to this uncertainty (Cubison and Jimenez, 2015). As
proton transfer mostly happens with acids, and nearly all HOM molecules will be charged by adduct
formation it is possible to infer the uncharged formula; therefore all HOMs from here onwards will
be listed as their uncharged form.

## 162 2.3. Size Distribution Measurements

Two Scanning Mobility Particle Sizer (SMPS) instruments measured particle size distributions at
15 minute time resolution, one LongSMPS (TSI 3080 EC, 3082 Long DMA, 3775 CPC, TSI, USA)
and one NanoSMPS (3082 EC, 3082 Nano DMA, 3776 CPC, TSI, USA) measuring the ranges 4-65
nm and 14-615 nm respectively. A Particle Size Magnifier (A10, Airmodus, FN) linked to a CPC
(3775, TSI, USA) measured the sub-3 nm size fraction. The PSM was run in stepping mode,
operating at four different saturator pressures to vary the lowest size cut-off of particles that it will
grow (this cut-off is technically a point of 50% detection efficiency) of <1.30, 1.36, 1.67 and 2.01
nm. The instrument switched between saturator pressures per 2.5 minutes, giving a sub-2.01 nm
size distribution every 10 minutes. The data was treated with a moving average filter to account for
jumps in total particle count, and due to the similar behaviour of the two upper and two lower size
cuts, these have been averaged to two size cuts at 1.30 and 1.84 nm.



## 2.4. Calculations

The condensation sink (CS) was calculated from the size distribution data as follows:

$$CS = 4\pi D \sum_{d'_p} \beta_{m,d'_p} d'_p N_{d'_p} \qquad (1)$$

where D is the diffusion coefficient of the diffusing vapour (assumed sulfuric acid), and $\beta_m$ is a

transition regime correction (Kulmala et al., 2012).

## 2.5. Other Measurements

$SO_2$ was measured using a 43i $SO_2$ analyser (ThermoFisher Scientific, USA), $O_3$ with a 49i $O_3$

analyser (ThermoFisher Scientific, USA) and $NO_x$ with a 42i-TL Trace $NO_x$ analyser

(ThermoFisher Scientific, USA), and a T500U CAPS $NO_2$ analyser (Enviro Technology Services,

USA). VOC mixing ratios were measured using a Proton Transfer Reaction-Time of Flight-Mass

Spectrometer (PTR-ToF 2000,Ionicon, Austria). Measurements of $J(O^1D)$ were carried out via filter

radiometers (Bohn et al., 2016) and measurements of OH, $RO_2$, and $HO_2$ concentrations were

carried out by Fluorescence Assay by Gas Expansion (FAGE) (Winiberg et al., 2016).

## 3. RESULTS AND DISCUSSION

## 3.1. Characteristics of Sampling Period

A total of five days of CI-API-ToF data were collected successfully, from 2017/06/21 midday

through 2017/06/26 midday. New particle formation events were observed on 24[th] June in the late

afternoon and 25th June at midday. Some nighttime formation of molecular clusters was seen

earlier in the campaign, as were several peaks to the 1.5 – 100 nm size range, likely from pollutant

plumes containing freshly nucleating condensable materials. The trace gases, $O_3$, $SO_2$, NO and $NO_2$

are plotted in the Figure S1. $O_3$ shows mid-afternoon peaks, around ~120 ppbv on the first two days

of the campaign, and 50-70 ppbv for the latter days. $SO_2$ shows a large peak, reaching 4 ppbv on

22/06 but <1 ppbv for the rest of campaign. NO shows strong mid-morning rush hour related peaks,
declining towards midday due to being rapidly consumed by $O_3$. $NO_2$ shows large traffic related
peaks. The sulfuric acid signal across this period as measured by $NO_3^-$ CI-APi-ToF showed strong
midday peaks, with concentrations highest on 24/06/2017 and 25/06/2017. The meteorological data
are shown in Figure S2 alongside condensation sink (CS). The conditions were generally warm and
humid, with temperature reaching its maximum on 25/06/2017, with a peak hourly temperature of
31°C. High temperatures were seen on 21/06 and 24/06 also, of 30°C and 26°C respectively.

**3.2.      Gas Phase HOM Chemistry**
**3.2.1.    Bulk chemical properties**
For the peaks that that have had chemical formulae assigned, oxidation state of carbon, or $OS_c$, can
be used to describe their bulk oxidation chemistry. $OS_c$ is defined as (Kroll et al., 2011)

$$OSc = (2 \times O:C) - H:C \quad (2)$$


This does not account for the presence of nitrate ester groups, which has been accounted for
previously by subtracting five times the N:C ratio (Massoli et al., 2018), under the assumption that
all nitrogen containing functionality is in the form of nitrate ester ($RONO_2$) groups. In Beijing,
multiple sources of nitrate-containing organic compounds are seen, in the forms of amines, nitriles
and heterocycles. The variation of oxidation state with carbon number ($C_n$) is plotted in Figure 1.
The average oxidation state of carbon in this dataset tends to decrease with an increase to $C_n$,
highest where $C_n = 5$, attributable both to high O:C and peak area for the peak assigned to
$C_5H_{10}N_2O_8$ at $m/Q$ 288. $C_n= 5$ also shows the greatest distribution of oxidation states, likely due to
the high ambient concentration of isoprene and therefore its many oxidation products being of high
enough concentrations for many well resolved peaks to be seen in this dataset. $C_n=10$ and 15 also
see a small increase to average oxidation number compared to their neighbours. The lower



oxidation state of the larger products is likely a function of two things. First and foremost, any
autoxidation mechanism must undergo more steps in order for a larger molecule to reach an
equivalent O:C ratio with a smaller one, and the equivalent O:C ratio is ultimately less likely to be
reached before the radical is terminated (Massoli et al., 2018). Secondly, the lower vapour pressures
of these larger products will lead to their partitioning into the condensed phase more readily than
the smaller, thus they are more rapidly lost (Mutzel et al., 2015).

The degrees of OSc observed here are similar to those seen in other environments such as during
the SOAS campaign in 2013 in southern United States, characterised by low $NO/NO_2$ and high
temperatures, where campaign averages of 0.3 ppb, 0.4-0.5 ppb, and 25°C respectively were
measured, although an additional parameter to account for nitrogen containing VOCs is included in
the calculation (Massoli et al., 2018). The $OS_c$ observed in Beijing is also higher than that seen in
the boreal forest environment of Hyytiälä, despite extremely low $NO_x$ concentrations, likely due to
low temperature conditions dominating in those conditions (Schobesberger et al., 2013). These
relatively similar degrees of oxidation to those seen in other, cleaner, environments are likely due to
an interplay between the higher temperatures facilitating rapid hydrogen abstractions (Crounse et
al., 2013; Praske et al., 2018; Quéléver et al., 2018) and the higher concentrations of $NO_x$, $HO_2^{\cdot}$, and
other $RO_2^{\cdot}$ molecules terminating the autoxidation sequence more efficiently.

A mass defect plot is shown in Figure 2.  The band of lower mass defect is characterised by a
number of large peaks with high signal, for example, at $m/Q$ 344 the ion $(NH_3)_3(H_2SO_4)_2HSO_4^-$ and
$(H_2O)_2(NH_3)_2(H_2SO_4)_2HSO_4^-$ at m/Q 362. Many water clusters are seen here. This clustering may
happen in the atmosphere, in the chemical ionisation inlet or through the critical orifice in the small
segmented quadrupole (SSQ) section of the instrument, and there is a weak dependence of these
concentrations on the SSQ pressure. The upper component of the mass defect is dominated by
organics, the upper end of more positive mass defect is occupied by molecules with more [1]H (mass





defect 7.825 mDa) and $^{14}$N (mass defect 3.074 mDa). The end of less positive mass defect has lower
$^{1}$H and more $^{16}$O (mass defect -5.085 mDa); alternatively put, the mass defect reflects the variation
in $OS_c$. The organic components with more positive mass defects will be more volatile than their
lower mass defect counterparts as they will contain fewer oxygen functionalities (Tröstl et al., 2016,
Stolzenburg et al., 2018). These higher volatility products may still contribute to larger size particle
growth. The more negative mass defect components will be those of greater O:C and therefore
lower volatility, LVOCs, and the yet larger and more oxidised components, ELVOCs (Tröstl et al.,

258 2016).


### 3.2.2.    Diurnal trends of HOMs

Temporal trends of HOMs in the urban atmosphere can throw light upon their sources and
behaviour in the atmosphere. Most of the HOM species peak in the daytime. These species all
follow a similar diurnal trend, as shown in Figure 3. Both the concentrations of $O_3$ and OH$^.$ are high
during this period (although the nitrate chemical ionisation technique is not sensitive to all OH$^.$
oxidation products (Berndt et al., 2015)). Figure S1 shows the time series of concentrations of NO
which is considered a dominant peroxy radical terminator of particular importance in the polluted
urban environment (Khan et al., 2015).  Peroxy radicals such as $HO_2$ and $RO_2$ also typically peak
during daytime.   The HOM components peaking in the daytime are presumed to be the oxidation
products of a mixture of anthropogenic and biogenic components, such as alkylbenzenes,
monoterpenes and isoprene. The oxidation of monoterpenes, specifically the monoterpene $\alpha$-pinene,
has been the subject of extensive study recently, with the $O_3$-initiated autoxidation sequence being
the best characterised (Ehn et al., 2014; Jokinen et al., 2014; Kurtén et al., 2015; Kirkby et al.,
2016); ozonolysis of $\alpha$-pinene opens the ring structure and produces a RO$^.{}_2$ radical (Kirkby et al.,
2016). In the case of aromatics, OH$^.$ addition to the ring and the subsequently formed bicyclic
peroxy radical is the basis for the autooxidation of compounds such as xylenes and
trimethylbenzenes (Molteni et al., 2018; Wu et al., 2017).



The identified compounds have been roughly separated into several categories, each of these plotted
in Figure 3. The top of this graph shows the separation of components into HOM and ON
(organonitrate) components. The ON signal is much higher than that of the HOM, attributable in
part to a few ions of high signal, such as the isoprene organonitrate $C_5H_{10}N_2O_8$. A few similar
structural formulae are seen ($C_5H_{10}N_2O_6$, $C_5H_{11}NO_6$, $C_5H_{11}NO_7$, etc), some of which have been
identified as important gas phase oxidation products of isoprene under high $NO_x$ conditions (Xiong
et al., 2015), and their contribution to SOA has been explored previously (Lee et al., 2016). A high
nitrophenol signal is also seen, $C_6H_5NO_3$. The signal for HOM compounds is less dominated by a
few large ions. The prevalence of ON compounds points towards the important role of $NO_x$ as a
peroxy radical terminator, with the probability for the $RO_2 + NO_x$ reaction to produce nitrate ester
compounds increasing with the size of the $RO_2$ molecule (Atkinson et al., 1982). The $NO_x$
concentrations in urban Beijing are approximately a factor of 10 higher than seen at the Hyytiälä
station in Finland as reported by Yan et al. (2016), and hence it is expected to be a more significant
peroxy radical terminator.

Despite the very large fluxes of anthropogenic organic pollutants in Beijing, biogenic emissions are
still an important source of reactive VOCs in the city, with abundant isoprene oxidation products
observed (see above), as well as monoterpene monomers ($C_{10}H_{16}O_9$, $C_{10}H_{15}O_9N$) and some dimer
products ($C_{20}H_{30}O_{11}$, $C_{20}H_{31}O_{11}N$). The time series of the concentrations all $C_5$, $C_{10}$ and $C_{20}$
molecules is plotted in the middle panel of Figure 3, with $C_5$ species assumed to be isoprene
dominated, $C_{10}$ and $C_{20}$ assumed to be monoterpene dominated. Isoprene oxidation products are
present at higher concentrations, with abundant isoprene nitrate and dinitrate products. $C_{10}$ products
show similar behaviour, with, for example, several $C_{10}H_{15}O_xN$ $x = 5-9$ compounds seen. The $C_{20}$
products seen are low in concentration, and follow the general formula $C_{20}H_xO_yN_z$, where $x =$
26−32, $y = 7-11$ and $z = 0-2$; in Figure 3 the signal for $C_{20}$ compounds has been multiplied by a





factor of 50 for visibility. The low concentrations reflect the lack of $RO_2^.$ cross reactions necessary
for the production of these accretion products.

Other identified peaks are plotted in the bottom panel of Figure 3. The $C_2$-$C_4$ components are
summed together, these being small organic acids such as malonic acid and oxalic acid, as well as
products such as $C_4H_7O_6N$. Malonic acid is the most prominent here, seen both as an $NO_3^-$ adduct
($C_3H_4O_4NO_3^-$) and a proton transfer product ($C_3H_3O_4^-$) at a ratio of around 2:3. The $C_6$-$C_9$
components are assumed to be dominated by oxidation products of alkylbenzenes such as $C_8H_{12}O_5$,
although fragments of other compounds, i.e., monoterpenes, can also occupy this region (Isaacman-
Vanwertz et al., 2018).  It is assumed the majority of the signal for these peaks come from
alkylbenzenes. This assumption is supported by the relative ratios of the monomer $C_8H_{12}O_n$
compounds being similar to those seen for xylene oxidation products in previous work (Molteni et
al., 2018). The largest fraction, $C_{11}$ through $C_{18}$, includes the larger compounds, oxidation products
of larger aromatics, or products of the cross reaction of smaller $RO_2^.$ radicals. Here they are grouped
without more sophisticated disaggregation as they all follow much the same time series, species
such as $C_{11}H_{11}O_8N$ following the same temporal trends as $C_{15}H_{16}O_9$ and $C_{16}H_{24}O_{12}$.

Nearly all ions with the exception of the larger compounds attributed to the cross reaction of $C_{10}$
monomers follow similar temporal patterns, with the majority of peaks occurring in the daytime.
This reflects the importance of the concentration of atmospheric oxidants. Some selected oxidation
products are plotted against their precursor VOCs in Figure 4. The concentration of isoprene is
plotted against the concentration of a nitrate HOM product, $C_5H_9NO_6$ (Xiong et al., 2015; Lee et al.,
2016), while monoterpenes are plotted against $C_{10}H_{16}O_9$ (Ehn et al., 2014; Berndt et al., 2016; Yan
et al., 2016; Kirkby et al., 2016; Massoli et al., 2018), and $C_2$-benzenes against $C_8H_{12}O_6$ (Molteni et
al., 2018; Wang et al., 2017). The first half of the time series shows little correlation between the
VOC species and the resultant oxidation products, while isoprene, monoterpenes and $C_2$-benzenes





follow their usual diurnal cycles, isoprene having the most distinct with a strong midday peak. The
latter two days, however, show similar and coinciding peaks in both the VOCs and HOMs - HOMs
show afternoon peaks on both days, and an initial shelf on the final half day. The $C_5H_9NO_6$ peak
follows some of the peaks of the isoprene, but not all (e.g., morning shelf of isoprene on 24/06).
Concentrations of isoprene do not seem to determine directly the concentration of HOM, as the day
with the lowest isoprene of all is the day with highest $C_5H_9NO_6$. The $C_{10}H_{16}O_9$ trace has
coincidental peaks with the monoterpene trace also, including two 4-hour separated simultaneous
peaks on 25/06. The peaks in the concentrations of $C_2$-benzenes are nearly synchronous with the
peaks in $C_8H_{12}O_6$; these exhibit a strong early afternoon peak likely due to the lack of an efficient
ozonolysis reaction pathway; the main oxidant of $C_2$-benzenes is the OH$^.$ radical. This behaviour is
much the same as the $C_3$-benzenes and their oxidation products. The concentration of precursor
VOC is likely a driving force in the identity and quantity of various HOM products, but not the sole
determinant, as while there are simultaneous peaks of VOCs and HOMs, both the condensation sink
and oxidant concentrations also influence HOM product concentrations.

The first half of campaign measurements is marked by an episode of low HOM concentrations. A
diurnal cycle still exists but it is weak. $J(O^1D)$ is used in Figure S1 as a proxy for radiation intensity,
and the radiation intensity is significantly lower on these prior days than it is on the 24th. No data is
available for the final period of measurement. Ozone is higher on the prior measurement days with
lower HOM concentrations (see Figure S2). It is therefore plausible that light intensity, and
therefore OH$^.$ concentration is one of the main drivers of HOM concentrations in Beijing.

The $C_{20}$ compounds show no strong diurnal sequence, contrasting with other HOMs. We can
presume that all $C_{20}$ compounds identified are the result of the reaction of two monoterpenoid $C_{10}$
RO$^.$$_2$ radicals, a reasonable assumption as all identified $C_{20}$ species follow the general formula
outlined for these reactions ($C_{20}H_{28-32}O_{6-16}$). The formation of $C_{20}$ dimers is dependent upon two



processes, initial oxidation of monoterpenes, and $RO_2-RO_2$ termination. Initial oxidation is
contingent upon oxidant concentration, which is highest in the daytime, and $RO_2\text{-}RO_2$ termination
is contingent upon the probability of the molecular collision between the $RO_2$ molecules occurring
before other radical termination (i.e., $RO_2-NO_x$, or $RO_2\text{-}HO_2$. There is likely a strong diurnal
sequence in the dominant $RO_2$ termination mechanisms across the day period, and the combination
of the two factors discussed above results in there being no strong diurnal trend in these molecules.
A lower oxidant concentration at night results in less $RO_2$ molecules, but less NO and $HO_2$ results
in a greater chance for those $RO_2$ molecules to dimerize. As the levels of $NO_x$ in Beijing fall, the
peroxy radical termination reactions will be less probable compared to continued autoxidation
(Praske et al., 2018), and it is expected that more oxidised HOM products will be seen with lower
volatilities and therefore a greater potential contribution to earlier stage particle formation and
growth.

### 3.3. New Particle Formation

Nearly all the signal intensity in the CI-APi-ToF instrument arises from molecules charged by $NO_3^-$,
therefore plotting the unit mass data against time describes simply the evolution of oxidised organic
molecules, acids and their molecular clusters both with each other and stabilising amine species.
This is done in Figure 5. As the signal intensity varies by factors of 10 from mass to mass, all
masses have been normalised to 1. This has been done separately for two days for clarity, as the
signal intensity also varies from day to day. PSM data for these two days is plotted in Figure 5 also,
with both total particle count >1.30 nm in black and the number difference between the lower and
upper size cuts (1.30 and 1.84 nm) in blue, which shows the number of particles between these
sizes. The relationship between mass and electrical mobility diameter can be defined thus (Tammet,

377 1995),


$$d_e = \left(\frac{6m}{\pi\rho}\right)^{\frac{1}{3}} + d_g \qquad (3)$$






where $d_e$ is the electrical mobility diameter of the cluster or particle, m is the mass of the cluster or
particle expressed in kg, $\rho$ is the density and $d_g$ is the effective gas diameter, determined to be 0.3
nm for smaller particles (Larriba et al., 2011). We can use this to draw a comparison between the
PSM and CI-APi-ToF measurements. If a density of 1.2 g cm$^{-3}$ is assumed, then once molecular
clusters reach the >400 $m/Q$ range, they will be seen in the lowest size cut of the PSM, or >700 $m/Q$
if a density of 2.0 g cm$^{-3}$ is assumed. A full table of densities is provided in the Supplementary
Information.

A burst in the signal seen by the CI-APi-TOF occurs first in the late morning in the top panel of
Figure 5, and this is at the same time as peaks begin to rise in the identified HOMs (see Figure 3).
Here, the PSM is not available due to an instrumental fault until 16:00; however, at that point, an
elevation to particle count and a large elevation to cluster count can be seen. Moving into the
evening period, the mass contour shows peaks to larger masses >400 $m/Q$. This is likely dimerized
compounds and $NO_3^-$ chemistry with little contribution to newly forming particles, but still sensitive
to chemical ionisation by $NO_3^-$. Many of these peaks cannot be assigned due to uncertainties in the
structural formula assignment for higher mass peaks, as the number of possible dimerised
compounds is many, being the combination of most possible $RO_2$ radicals. Graphically, these are
over-represented in Figure 5 due to the normalisation, their concentrations (especially >500 $m/Q$)
are much lower than the concentrations <400 $m/Q$.

The second day plotted in the lower panel of Figure 5 (25/06/2017) shows a strong afternoon peak
to the HOMs (for most HOMs, stronger than that on the day prior). Particle formation is shown in
the PSM data. A strong midday peak to particle number is seen with two distinct peaks to cluster
count. These two peaks are not coincidental with the two peaks to HOM concentrations (Figure 3)
nor the two peaks in aromatic VOCs (i.e., $C_2$-benzenes in Figure 4). Sulfuric acid, however, does



peak synchronously with the particle number count. Sulfuric acid is plotted across the contour plot
in Figure 6, where PSM data is also shown in the bottom panel. The peak to CI-APi-TOF mass,
visible in Figure 5 occurs at around 12:00/13:00, peaks in the PSM cluster count occur at 10:00 and
13:00 - at 13:00 the peaks in mass occur between 200-550 $m/Q$. Assuming the density of the
identified species is ≤1.6 g cm$^{-3}$ then these will be suitably sized to be grown in the PSM saturator
above the size cut at 1.30 nm. The peak at 10:00 in PSM cluster count is characterised by a few
peaks at specific masses (around 680, 720, 840, 860 $m/Q$), presumably specific nucleating inorganic
clusters, pointing towards a possible evolution in the composition of clusters throughout the
nucleation event with the early nucleation linked with a few specific precursors. These newly
formed particles then go on to grow and contribute significantly to the larger particle count (Figure
S3). As initial particle formation coincides with sulfuric acid concentrations and before HOM
concentrations peak, it can be assumed on these days, the HOM contribution to the initial particle
formation is modest.

There is recent strong evidence to suggest that the driving force of the earliest stages of particle
formation in urban Shanghai is from sulfuric acid and $C_2$-amines (Yao et al., 2018), supported by
the coincidental peaks of sulfuric acid with new particles as seen in Figure 6. Dimethylamine
(DMA) can efficiently stabilise the sulfuric acid clusters (Almeida et al., 2013). Here, few larger
sulfuric acid-DMA clusters were visible in the dataset, as seen in the work by Yao et al., 2018,
although five SA-DMA ions were observed, the others were likely too low in signal to be
confidently resolved from their neighbouring peaks. The scarcity of sulfuric acid-DMA clusters is
likely due to instrumental conditions, rather than their absence in the atmosphere. The nitrate
chemical ionisation system tends to evaporate amine compounds upon charging, and as specific
voltage-tuning setups can lend themselves towards preservation or breakage of molecular clusters,
the signal for larger sulfuric acid clusters was also very weak. The formation of HOM-sulfuric acid
clusters is unlikely under atmospheric conditions (Elm et al., 2017) and few of these were observed.



Concentrations of HOMs seem to coincide with later particle growth; it can be expected that HOM
molecules make a more significant contribution to particle growth than to early particle formation,
with the largest and most oxidised being involved in early growth, and the smaller and less oxidised
contributing to later growth as the necessary vapour pressure properties become less demanding.

**4.      CONCLUSIONS**
The average degree of HOM oxidation in Beijing is comparable with that seen in other
environments. Rapid intramolecular hydrogen shifts during autoxidation due to the higher
temperatures are probably offset by the frequent termination reactions due to high $NO_x$
concentrations. $OS_c$ values seem to be marginally higher for biogenic species.

The temporal trend of nearly every HOM shows a daytime maximum. Both $O_3$ and $OH^.$ have high
daytime concentrations and these likely drive the initial oxidation steps. The species arising from
alkylbenzene precursors show sharper afternoon peaks, probably since their oxidation is $OH^.$
dominated. Many of the rest of the peaks, coming from largely BVOC precursors show broader
daytime peaks, being influenced by $O_3$ also. There seems to be no direct link between VOC
concentrations and HOM concentrations, with days of lower precursor VOC sometimes having
higher HOM concentrations and vice versa.

Initial particle formation coincides with peak sulfuric acid concentrations, while the growth of the
particles correlates more closely with the concentrations of HOMs. This is very similar to behaviour
observed in a study of NPF in Shanghai which was attributed to sulphuric acid-dimethylamine-
water nucleation with condensing organic species contributing to particle growth (Yao et al., 2018).
The freshly formed particles grow and contribute significantly to total particle loading. This is
visible when the unit mass CI-APi-ToF data is plotted as a contour plot, and further to this is visible
in the PSM data, with bursts to both total number count >1.30 nm and the number of molecular



clusters between 1.30 and 1.84 nm. As $NO_x$ levels fall in Beijing due to traffic emission control measures being enforced it is likely that autoxidation will become increasingly significant in the new particle formation processes. The number of molecules detected by the $NO_3^-$ CIMS is undoubtedly many more than have had formulae assigned here, but to identify more requires a more sophisticated data deconvolution.

**DATA ACCESSIBILITY**

Data supporting this publication are openly available from the UBIRA eData repository at

https://doi.org/10.25500/edata.bham.00000304

**AUTHOR CONTRIBUTIONS**

The study was conceived and planned by RMH and ZS. DCSB and JB set up and operated the main instrumental measurements, and JB prepared the first draft of the paper and responded to comments from RMH and ZS. CNH and WJA contributed the hydrocarbon data and provided comments on the draft manuscript.

**COMPETING INTERESTS**

The authors have no conflict of interests.

**ACKNOWLEDGMENTS**

This work was part of the APHH-Beijing programme funded by the UK Natural Environmentl Research Council (NE/N007190/1) and the Natural Sciences Funding Council of China. It was additionally facilitated by the National Centre for Atmospheric Science ODA national capability programme ACREW (NE/R000034/1), which is supported by NERC and the GCRF. We thank Professor X.M Wang from the Guangzhou Institute of Geochemistry, Chinese Academy of Sciences, Brian Davison from Lancaster University and Ben Langford, Eiko Nemitz, Neil



Mullinger and other staff from the Centre for Ecology and Hydrology, Edinburgh for assistance
with the VOC measurements and associated infrastructure.





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



**FIGURE LEGENDS:**

**Figure 1**   Oxidation state of carbon calculated as two times the oxygen to carbon ratio minus the hydrogen to carbon ratio against carbon number for (colored) individual ions and (blue circles) signal weighted average for each carbon number. Area and colour are both proportional to the peak area for each ion

**Figure 2**   Kendrick mass defect plot of fitted mass spectral peaks between 200-700 mass units where carbon is the Kendrick base. Kendrick mass defect can be defined as the Kendrick integer mass - Kendrick mass. The size of point is proportional to the signal intensity. As 1H has a positive mass defect (1.007276 Da), the upward trend along the horizontal indicates increasing carbon chain length, and differences at similar masses are due to increasing oxygen functionality, clustering with species such as sulfuric acid (negative mass defect) and ammonia (positive mass defect), as 16O and 32S have negative mass defects (15.9949 and 31.9721 Da respectively), while 14N has a positive mass defect at 14.0031 Da. Here, two ions at 201 and 288 m/Q have been removed due to high signal.

**Figure 3**   Summed time series of the concentrations of (top) all non-nitrogen containing HOMs and all organonitrates identified, (middle) C5, C10 and C20 components, assumed to be dominated by isoprene, monoterpene monomer and monoterpene dimers, signal for C20 multiplied 50 times to fit scale, and (bottom) summed C6 - C9 components, and summed C11 - C18 components, assumed to be dominated by alkylbenzenes and other larger components respectively.

**Figure 4**   Time series for the whole sampling campaign for the concentrations of (left axis) VOCs as measured by PTR-ToF and (right axis) a selected HOM product associated with that precursor.

**Figure 5**   Normalised unit mass NO3- CI-APi-ToF signal intensity on 24/06/2017 (top) and 25/06/2017 (bottom). Each individual unit mass was normalised to a maximum of 1. Each period is normalised separately so the individual signal maxima on each day are visible. The graph is plotted between 200-600 mass units, with every 10 mass units averaged for simplicity. On the secondary axis is plotted PSM data, both total particle count >1.30 nm (black trace) and total clusters between 1.30 and 1.84 nm (blue trace). Data is plotted at 1 hour time resolution.

**Figure 6**   SMPS + PSM contour plot for two nucleation days on 24/06/2017 and 25/06/2017. Data in bottom panel is from the PSM instrument, top panel from NanoSMPS, units in colour bar are $\log_{10} (dN/\log D_p)$ for N in cm$^{-3}$. Points signify normalised sulfuric acid concentration (right axis) as measured by CI-APi-ToF.





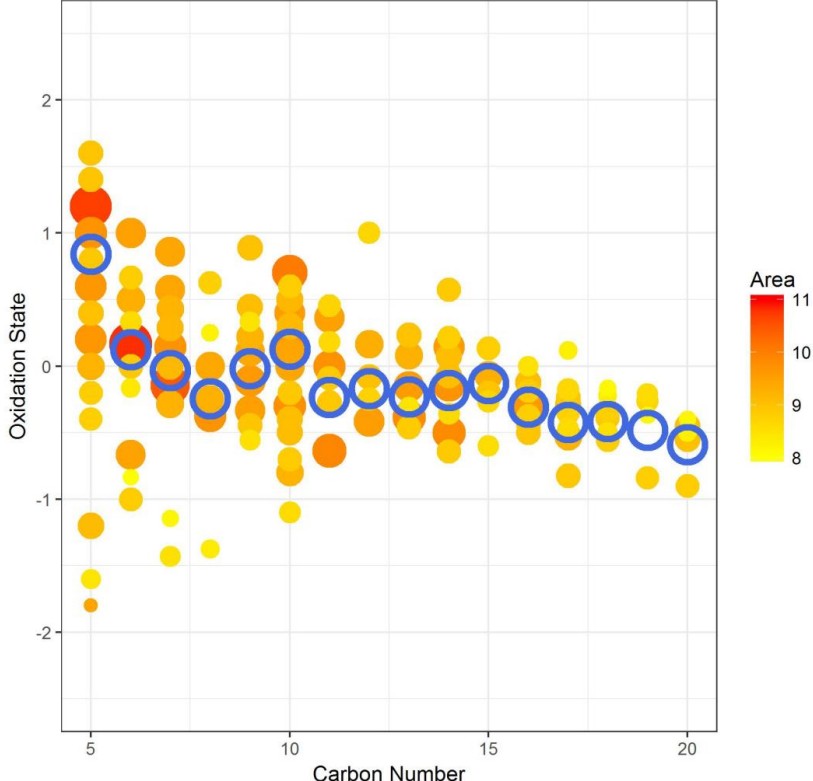


**Figure 1.** Oxidation state of carbon plotted against carbon number for (colored) individual ions and
(blue circles) signal weighted average for each carbon number. Area and colour are both proportional
to the peak area for each ion.





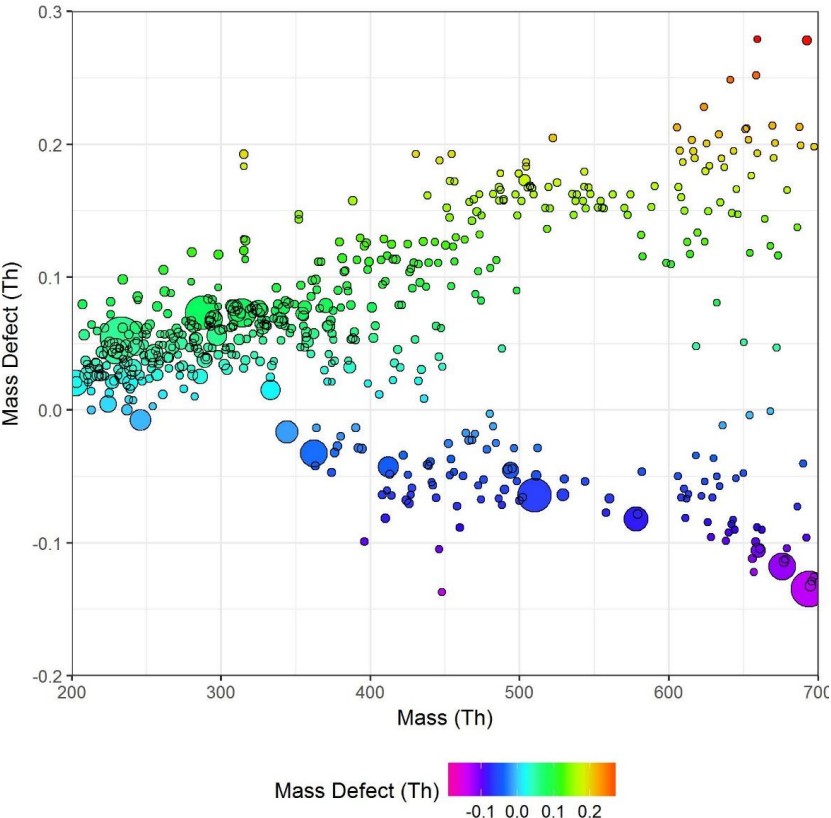


**Figure 2.** Kendrick mass defect plot of fitted mass spectral peaks between 200-700 mass units where carbon is the Kendrick base. Kendrick mass defect can be defined as the Kendrick integer mass - Kendrick mass. The size of point is proportional to the signal intensity. As [1]H has a positive mass defect (1.007276 Da), the upward trend along the horizontal indicates increasing carbon chain length, and differences at similar masses are due to increasing oxygen functionality, clustering with species such as sulfuric acid (negative mass defect) and ammonia (positive mass defect), as [16]O and [32]S have negative mass defects (15.9949 and 31.9721 Da respectively), while [14]N has a positive mass defect at 14.0031 Da. Here, two ions at 201 and 288 $m/Q$ have been removed due to high signal.





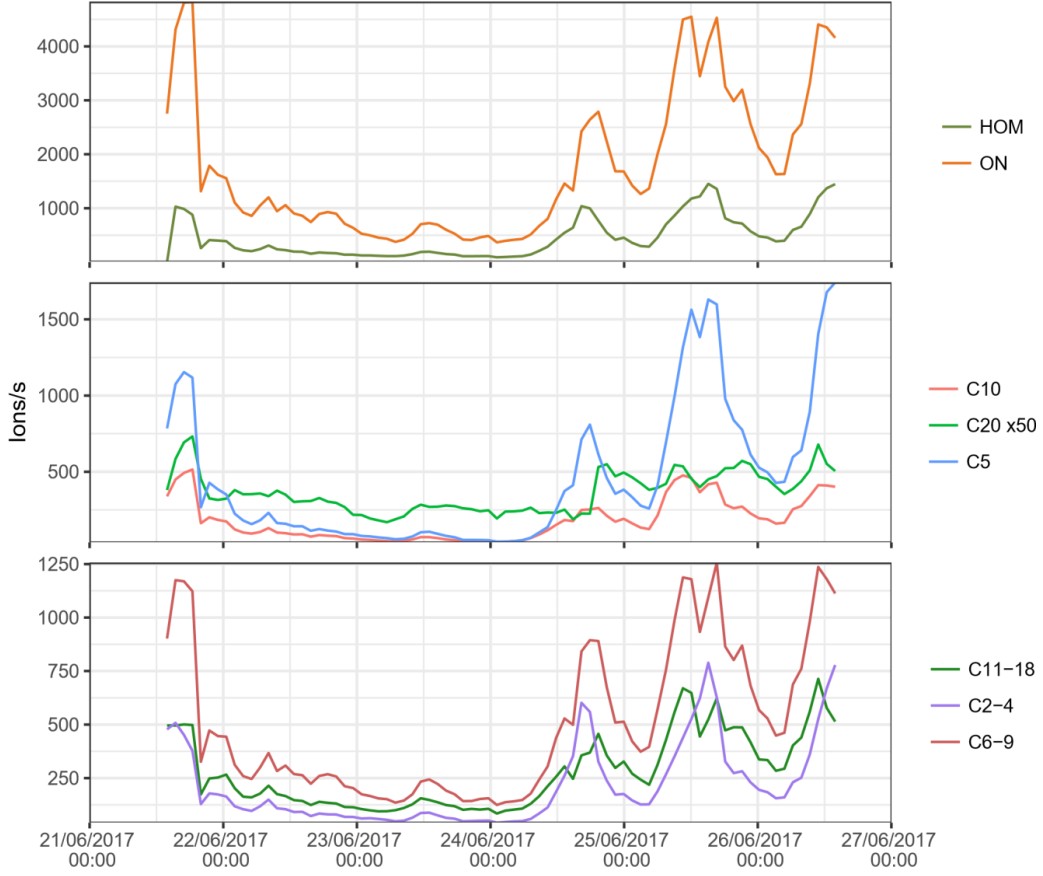

**Figure 3.** Summed time series of the concentrations of (top) all non-nitrogen containing HOMs and all organonitrates identified, (middle) C5, C10 and C20 components, assumed to be dominated by isoprene, monoterpene monomer and monoterpene dimers, signal for C20 multiplied 50 times to fit scale, and (bottom) summed C6 - C9 components, and summed C11 - C18 components, assumed to be dominated by alkylbenzenes and other larger components respectively.



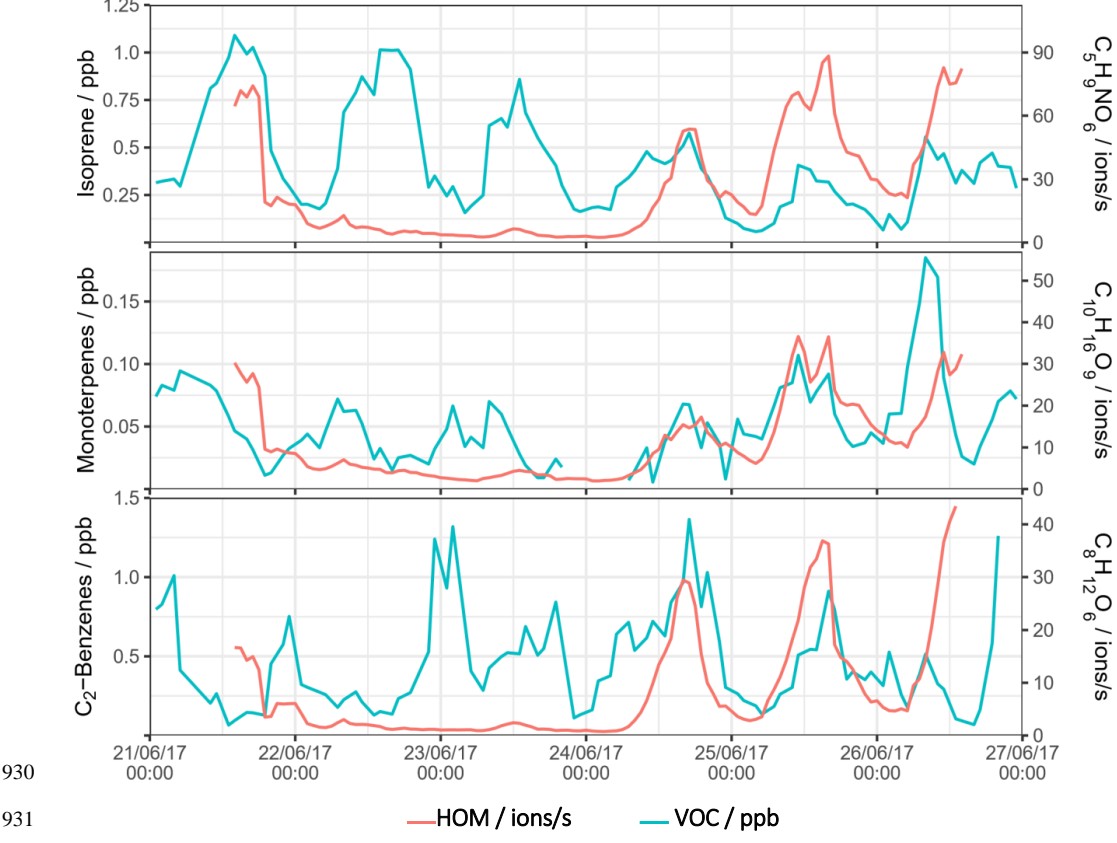

**Figure 4.** Time series for the whole sampling campaign for the concentrations of (left axis) VOCs as measured by PTR-ToF and (right axis) a selected HOM product associated with that precursor.





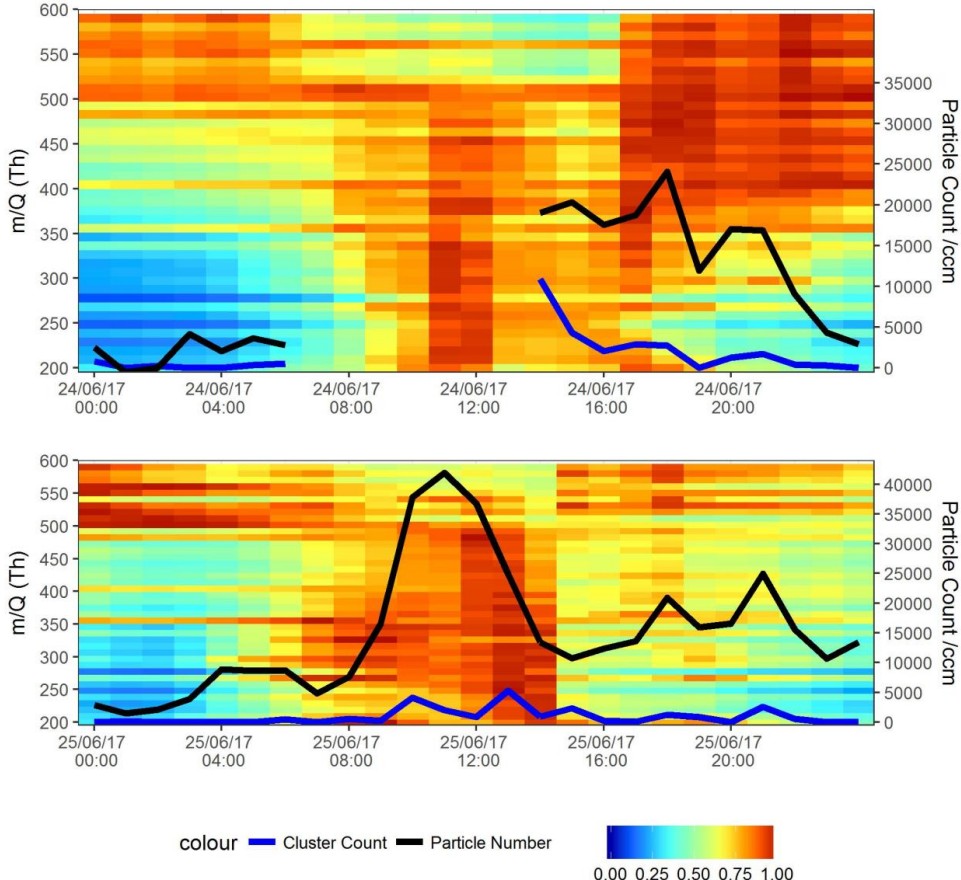

**Figure 5.** Normalised unit mass $NO_3^-$ CI-APi-ToF signal intensity on 24/06/2017 (top) and 25/06/2017 (bottom). Each individual unit mass was normalised to a maximum of 1. Each period is normalised separately so the individual signal maxima on each day are visible. The graph is plotted between 200-600 mass units, with every 10 mass units averaged for simplicity. On the secondary axis is plotted PSM data, both total particle count >1.30 nm (black trace) and total clusters between 1.30 and 1.84 nm (blue trace). Data is plotted at 1 hour time resolution.





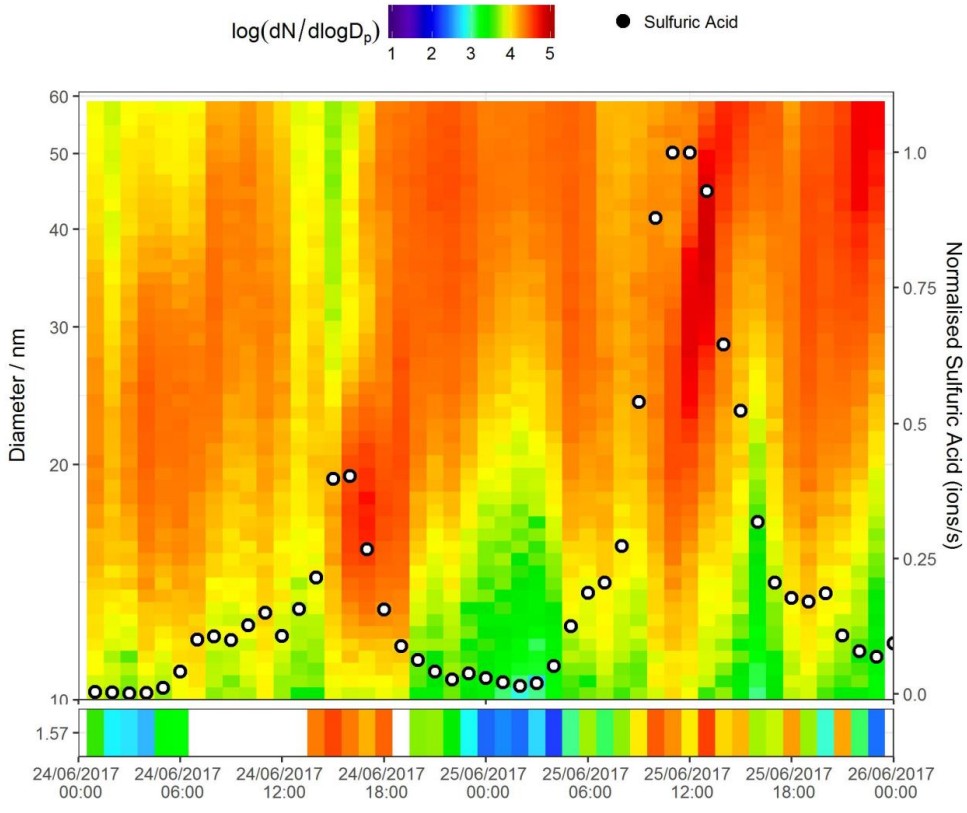

941

**Figure 6**. SMPS + PSM contour plot for two nucleation days on 24/06/2017 and 25/06/2017. Data in bottom panel is from the PSM instrument, top panel from NanoSMPS, units in colour bar are $\log_{10}$ $(dN/\log D_p)$ for N in cm$^{-3}$. Points signify normalised sulfuric acid concentration (right axis) as measured by CI-APi-ToF.




