# Peer review of "OBSERVATIONS OF HIGHLY OXIDISED MOLECULES AND PARTICLE NUCLEATION IN THE ATMOSPHERE OF BEIJING"

_Atmospheric Chemistry and Physics, 2019_

## Referee Comment (RC1) · Anonymous Referee #3 · 4 Jun 2019

The manuscript by Brean et al. describes measurements from Beijing during spring/summer 2017. The main instrument deployed was a nitrate chemical ionization atmospheric pressure interface time of flight mass spectrometer (CI-APi-TOF) for the measurement of sulfuric acid and highly-oxygenated organic molecules (HOM). Other measured parameters include the particle size distribution, cluster and nanoparticle concentrations, meteorological conditions and mixing ratios of certain trace gases ($SO_2$, $NO_x$, $O_3$). A proton transfer reaction mass spectrometer (PTR-MS) measured the mixing ratios of isoprene, monoterpenes and $C_2$-benzenes. The main focus of the paper lies on the description of the observed HOM signals and relating these to observed new particle formation events. One conclusion is that the occurrence of nucleating clusters correlates with peak sulfuric acid concentrations, whereas the peak HOM concentrations occur at a later time and are thus rather related to particle growth than to nucleation. The authors further speculate that dimethylamine (DMA) together with sulfuric acid (SA) could be responsible for nucleation due to the observation of some SA-DMA-containing species. The present study describes the HOM signals in a mega-city environment where both anthropogenic and biogenic emissions are relevant. It is therefore an important contribution because previous studies mainly focused on environments where biogenic emissions dominate (e.g., Hyytiälä Finland or the Southeastern United States). The manuscript is very well-written and structured. It is also very much appreciated that a full list of identified signals from the CI-APi-TOF mass spectra is provided in the supplementary information. One flaw of the present study is that no information on $HO_x$ and $RO_2$ is provided although the authors state that these compounds have been measured. Furthermore, I think that some more information can possibly be retrieved regarding the relevant nucleation mechanism. Suggestions for further data evaluation in this direction are provided below. Several further specific suggestions for improvements are listed in the following. These should be implemented before publication in ACP.

L27: please define the used acronyms (VOC, BVOC)

L27: It would be good to mention already in the abstract when the data were taken (month and year).

L37: "$O_3$ is lower on the days with higher HOM concentrations": This sounds as if $O_3$ inhibits the HOM formation. Can this just be coincidence as there are relatively few days of measurements?

L135: 3 sccm of carrier (sheath?) gas flow for $N_2$ is very low as this flow is typically on the order of 20 to 30 slm in CI-APi-TOF instruments, please check. In addition, only one unit for the flows should be used (currently Lpm, sccm and SLM are used).

L145: Usually the nitric acid trimer (m/z 188, i.e., $(HNO_3)_2NO_3^-$) yields a rather high signal in nitrate CI-APi-TOF spectra, too. If this signal is not observed it points to rather strong fragmentation of cluster ions. Is the trimer signal missing completely? Furthermore, it is mentioned here that all signals are normalized with the primary ion count rates; however, in the figures this normalization seems to be missing. The statement here also contradicts the statement in L149/150 ("… all values are reported in signal intensity, ions/s.").

L149/150: Rather than reporting signal intensity (ions/s) I highly recommend to report normalized signals in all figures, i.e., the data should be normalized by the sum of all primary ions (m/z 62, 80, 125 and 188, if present). It would also be good to mention that the conversion constant (from normalized counts to concentrations) is typically between a few $10^9$ and $1\times10^{10}$ molecule $cm^{-3}$ (see e.g., Kürten et al., 2012). In this way the reader can get an idea of the rough HOM and sulfuric acid concentrations. One further suggestions relates to the fact, that concentrations of $SO_2$ and OH were measured along with the condensation sink. From these data the $H_2SO_4$ concentration can be estimated (using a simple steady-state assumption for the main source and the sink of $H_2SO_4$). In this way, an estimate for the calibration constant can be derived.

L150: It would be good to mention typical values for the mass resolving power and mass accuracy.

L165/166: Please swap the order of the reported size ranges as the LongSMPS is mentioned before the NanoSMPS.

L168 and L170: The term "saturator pressures" is used here; however, in the PSM the saturator flow rates are varied in order to achieve different diethylene glycol supersaturations; this should be clarified.

L172: It is not clear what is meant by "similar behavior of the upper and two lower size cuts". Do the authors mean that the concentrations for the lower and upper two size channels typically correlate very well?

L187: It is mentioned that OH, $RO_2$ and $HO_2$ concentrations were measured, yet, none of these data are shown. To my knowledge the present study is the first ambient study where HOM, $O_3$, OH, $HO_2$ and $RO_2$ were measured simultaneously. Therefore, a lot could be learned about the different HOM formation pathways (e.g., if certain HOM originate rather from reactions with OH or $O_3$). It would be great if somehow the $HO_x$ data could be incorporated in the data analysis.

Figure S1: please show the (normalized, see comment above) $H_2SO_4$ signals on a log scale

L209: delete one of the "that"

L221: I think some of the signals cannot be unambiguously identified, e.g., the mentioned sum formula could also be written as $C_5H_8O_2(HNO_3)_2$ or $C_5H_9NO_5(HNO_3)$, where the $HNO_3$ could be coming from the charger ions (i.e., $(HNO_3)_2NO_3^-$ or $(HNO_3)NO_3^-$ rather than $NO_3^-$). One way to test this hypothesis is to check if the m/z 288 signals correlates with m/z 225 (this could be the same neutral molecule just with one less $HNO_3$ from the charging process). I also think that this possibility of ambiguity exists for some other nitrogen containing species, which affects the evaluation of the oxidation state values shown in Figure 1. Although the question of ambiguity cannot be ultimately resolved it should be mentioned and discussed briefly.

L245/246: Schobesberger et al. (2015) provide a detailed list of observed signals in the nucleating system of sulfuric acid and ammonia. From their observations prominent signals for

the reported masses (m/z 344 and m/z 362) seem rather unlikely. I would also be surprised if just these two mixed ammonia-sulfuric acid peaks show up in the spectra without any others. Have the authors considered the isotopic distributions of the assigned signals in their analysis? Sulfur has a distinct isotopic pattern; therefore, the assigned formulas in Table S2 for the sulfur-containing species could be checked by considering the isotopes.

L267/268: As mentioned before, it would be great if more information on $HO_x$ and $RO_2$ could be provided.

L295: the plot does not show concentrations but the raw signals

L344: J(O1D) is not shown in Figure S1

L347: neither $O_3$ nor HOM are shown in Figure S2

L410: in the PSM particles are grown within the condenser

L411 and L412: Can the authors at least speculate what compounds cause these signals? If they are from (in)organic compounds ($H_2O$, $NH_3$, $H_2SO_4$ and maybe amines) the number of possible combinations should not be too large.

L420 to 430: The possibility of sulfuric acid-amine nucleation should be further discussed. To me it seems very unlikely that only selected SA-DMA clusters show up in the spectra. For nitrate CI-APi-TOF measurements a detailed study of sulfuric acid-dimethylamine clusters has recently been presented (Kürten et al., 2014). That study has also shown that DMA together with sulfuric acid forms new particles very efficiently; therefore, tiny amounts (pptv) should suffice for efficient nucleation and the presence of DMA in clusters is already evidence that DMA is assisting in NPF. I suggest to search for further DMA (or other amine) containing clusters and to check if ambiguity can be ruled out, e.g., that the clusters with DMA and sulfuric acid are not due to some other (organic) compound. This can be done by taking into account the isotopic patterns. In addition, in Table S2 one of the listed clusters is $C_2H_7NHSO_4^-$ (i.e., a $C_2$-amine clustered with the bisulfate ion). This cluster does, however, not exist as the Lewis base ($HSO_4^-$) does not form a stable cluster with a strong base ($C_2$-amine) unless at least two further acids ($H_2SO_4$) are present in the cluster (Ortega et al., 2014; Kürten et al., 2014).

Figure 2: Is this MD plot corresponding to a period when NPF is occurring? It would be good to show a second MD plot for another day (same time of day) when no NPF is occurring just to see what signals could make the difference. In addition, there seem to be really prominent peaks (negative MD) at m/z of ~500 and ~700. Have the corresponding compounds been identified? Do these signals show a distinct diurnal pattern with higher concentrations during NPF?

**References:**

Kürten, A., et al.: Calibration of a chemical ionization mass spectrometer for the measurement of gaseous sulfuric acid, *J. Phys. Chem. A*, 116, 6375–6386, https://doi.org/10.1021/jp212123n, 2012.

Kürten, A., et al.: Neutral molecular cluster formation of sulfuric acid-dimethylamine observed in real-time under atmospheric conditions, *Proc. Natl. Acad. Sci. USA*, 111, 15019–15024, https://doi.org/10.1073/pnas.1404853111, 2014.

Ortega, I. K., et al.: Electrical charging changes the composition of sulfuric acid–ammonia/dimethylamine clusters, *Atmos. Chem. Phys.*, 14, 7995–8007, https://doi.org/10.5194/acp-14-7995-2014, 2014.

Schobesberger, S., et al.: On the composition of ammonia–sulfuric-acid ion clusters during aerosol particle formation, *Atmos. Chem. Phys.*, 15, 55–78, https://doi.org/10.5194/acp-15-55-2015, 2015.

---

## Referee Comment (RC2) · Anonymous Referee #2 · 19 Jun 2019

Access Review of the MS entitled "Observations of highly oxidised molecules and particle nucleation in the atmosphere of Beijing" by Brean et al.,

Authors' have used state-of-the-art Chemical ionization atmospheric pressure interface-time of flight mass spectrometer (CI-APi-TOF-MS) to characterize the HOMs as well as the particle size distributions using the Scanning mobility particle sizer (SMPS), other trace gases such as ozone and SO2 using respective commercial online gas-line analyzers from the Urban Beijing environment. The manuscript is within the scope of the ACP. The topic discussed here is the "role of highly oxidized multifunctional organic molecules (HOMs) from polluted Beijing" is most relevant in the present day context of increase in human activities and associated emission of VOCs, their transformation in the atmosphere as these contribute to ambient particulate load

and, thereby, can have considerable health-effects. All the discussions about the role of HOMs on particle growth followed by NPF events due sulfuric acid clusters in Urban Beijing are well supported by the observations. I recommend publication of the manuscript after revision. Some comments below could improve the readability of the manuscript.

L 26: VOC abbreviation in the abstract is not defined L28-31: sentence is hard to read L 58. Delete 'the' before 'many' L76: Add a ", the" after compound L77: Abbreviate BVOC here. Biogenic should compare with anthropogenic VOCs. So correct these statements. L87-88: Statement is not clear. L129: change "organics" to "organic compounds" L111: what is APHH here? L179: what is it stand for d'p and Nd'p here? L186: What is J(O'D)?? Please provide some baseline information here. L244-258: This text is not clear enough to understand the Figure 2. Some background information should be provided how to interpret the data and what is observed followed by what does it mean? L261: change 'throw light upon' with 'reveal' L263: peak in the daytime? But it looks from Figure 3 that mostly peaking in the evening/night time. Why there is no peak on 23-24/06/2017 L268: HOM components peaking in the daytime? From Figure 3, it is not clear. What is the basis for this assumption that HOMs are produced by the oxidation of anthropogenic/biogenic components (e.g., alkylbenzenes, monoterpenes, isoprene). Figure 3: C6 - C9 components, and summed C11 - C18 components, assumed to be dominated by alkylbenzenes and other larger components respectively-how this was assumed? L 284: Please add panels 'a, b, and c' in Figure 3 and accordingly refer these in the discussion in the manuscript here. L312: Relative to what?? L320: majority of peaks occurring the daytime? But from Figure 3, it is mostly in evening time. L336: Early afternoon peak? By looking at Figure 4, it looks like evening hrs. The scale showed '0:00 HRS' – Is it 24:00 hr? L337-338: From where, it is inferred this (i.e., similar behaviour of C3-benzenes and their oxidation products as C2-benzenes and their HOMs)? L344: I could not find J(O'D) in Figure S1! L346-347: Figure S2 does not provide this information, please double check and maintain consistency between text and supplementary information. L350: This is inferred from which figure, please mention. L369: text is unclear-'what is unit mass data' L371-372: This normalization part is not clear enough to follow the figure 5. Please elaborate. L403-404: sentence is not clear. In Figure 3, point the two peak of HOMs on 25/06/2017 to understand the text described here. L405: I am not able to see the two peaks in C2-benzenes in Figure 4. Please encircle or write clearly to maintain consistency with figure. L407-9: These sentences are not clear. Please consider rephrasing these sentences L412: From Figure 5, PSM cluster peaked at 10:00 and 13:00 h have m/Q between 200-550 (as also stated on L409). But the specified m/Q here is beyond the scale shown in Figure 5. Is it correct or I am missing something. L412: Add 'because of' after 'presumably' L416-418: Please refer to the figure. L425: Define 'SA-DMA' here. L443: From the Figure 3, HOM peaked in the evening hours on 24/06/2017 compared to 25/06/2017, where HOM peaked at the early afternoon. So 'daytime peak of HOMs' need to be rephrased.

---

## Author Comment (AC1) · 23 Jul 2019

Journal: ACP MS No.: acp-2019-156 MS Type: Research article Title: Observations of Highly Oxidised Molecules and Particle Nucleation in the Atmosphere of Beijing Author(s): James Brean et al.

RESPONSE TO REVIEWERS We thank the reviewers for their very helpful comments and are pleased to respond.

REVIEWER #1 L27: please define the used acronyms (VOC, BVOC) RESPONSE: Definitions added. L27: It would be good to mention already in the abstract when the data were taken (month and year). RESPONSE: Dates added. L37: "O3 is lower on the days with higher HOM concentrations": This sounds as if O3 inhibits the HOM formation. Can this just be coincidence as there are relatively few days of measurements? RESPONSE: This is probably not O3 inhibiting the formation, this just indicates that O3 may not be as important an oxidant as OH.. The wording here has been changed to better indicate this. 135: 3 sccm of carrier (sheath?) gas flow for N2 is very low as this flow is typically on the order of 20 to 30 slm in CI-APi-TOF instruments, please check. In addition, only one unit for the flows should be used (currently Lpm, sccm and SLM are used). RESPONSE: Carrier flow refers to the small flow of N2 across the surface of liquid HNO3, carrying HNO3 through to the inlet to produce NO3-. This has been reworded for clarity and the rest of the units have been fixed. L145: Usually the nitric acid trimer (m/z 188, i.e., (HNO3)2NO3 ‒ ) yields a rather high signal in nitrate CI-APi-TOF spectra, too. If this signal is not observed it points to rather strong fragmentation of cluster ions. Is the trimer signal missing completely? Furthermore, it is mentioned here that all signals are normalized with the primary ion count rates; however, in the figures this normalization seems to be missing. The statement here also contradicts the statement in RESPONSE: The trimer signal is present in these spectra, just relatively small compared to these other reagent ion peaks, so would make a relatively small change to this normalization, and there are other occasional peaks which appear within one full-width-half-maximum of the peak at 188, causing some uncertainty in the signal intensity. L149/150 ("... all values are reported in signal intensity, ions/s.").Rather than reporting signal intensity (ions/s) I highly recommend to report normalized signals in all figures, i.e., the data should be normalized by the sum of all primary ions (m/z 62, 80, 125 and 188, if present). It would also be good to mention that the conversion constant (from normalized counts to concentrations) is typically between a few 109 and 1×1010 molecule cm-3 (see e.g., Kürten et al., 2012). In this way the reader can get an idea of the rough HOM and sulfuric acid concentrations. One further suggestions relates to the fact, that concentrations of SO2 and OH were measured along with the condensation sink. From these data the H2SO4 concentration can be estimated (using a simple steady-state assumption for the main source and the sink of H2SO4). In this way, an estimate for the calibration constant can be derived. RESPONSE: This

was a mislabelling of the axes. All of these signals have been normalised to reagent ion counts of 1e+5 and this has been fixed on all figures. Unfortunately the OH, HO2 and RO2 data have very little temporal overlap with our own (discussed below) so the resultant H2SO4 proxy had very little crossover with our own values. Values of the calibration constant have been calculated from the very limited data available and are now included.

L150: It would be good to mention typical values for the mass resolving power and mass accuracy. RESPONSE: These have been added (3500 m/dm, 20ppm @ 288 m/Q). L165/166: Please swap the order of the reported size ranges as the LongSMPS is mentioned before the NanoSMPS. RESPONSE: Fixed. L168 and L170: The term "saturator pressures" is used here; however, in the PSM the saturator flow rates are varied in order to achieve different diethylene glycol supersaturations; this should be clarified. RESPONSE: Fixed. This should have read saturator flows. L172: It is not clear what is meant by "similar behavior of the upper and two lower size cuts". Do the authors mean that the concentrations for the lower and upper two size channels typically correlate very well? RESPONSE: Each member of the two smaller (<1.3 and 1.36 nm) and two larger (1.67 and 2.01 nm) correlated well and also provided data of near identical magnitudes, so the average of these were taken to produce just one single dN/dlogdp value. L187: It is mentioned that OH, RO2 and HO2 concentrations were measured, yet, none of these data are shown. To my knowledge the present study is the first ambient study where HOM, O3, OH, HO2 and RO2 were measured simultaneously. Therefore, a lot could be learned about the different HOM formation pathways (e.g., if certain HOM originate rather from reactions with OH or O3). It would be great if somehow the HOx data could be incorporated in the data analysis. RESPONSE: Unfortunately the FAGE data was only coincidental with a small amount of the CIMS data. There is about 19 hours of overlapping data on 21/06/2017, and a few hours on 23/06 and overnight on 24/06/2017 – 25/06/2017. As this data is sparse, we felt it was not enough to add any meaningful interpretation of our own data. Figure S1: please show the (normalized, see comment above) H2SO4 signals on a log scale

RESPONSE: This has been fixed. NO and NO2 are now also on log scales. L209: delete one of the "that" RESPONSE: Removed. L221: I think some of the signals cannot be unambiguously identified, e.g., the mentioned sum formula could also be written as C5H8O2(HNO3)2 or C5H9NO5(HNO3), where the HNO3 could be coming from the charger ions (i.e., (HNO3)2NO3 ‒ or (HNO3)NO3 ‒ rather than NO3 ‒ ). One way to test this hypothesis is to check if the m/z 288 signals correlates with m/z 225 (this could be the same neutral molecule just with one less HNO3 from the charging process). I also think that this possibility of ambiguity exists for some other nitrogen containing species, which affects the evaluation of the oxidation state values shown in Figure 1. Although the question of ambiguity cannot be ultimately resolved it should be mentioned and discussed briefly. RESPONSE: Checking all of these signals was part of our analyses. None of these peaks correlated with their nitrate monomer/dimer/trimer counterparts. If some of our formulae were to exist as clusters with the nitrate dimer, it would follow that their cluster with the nitrate monomer would be seen 63 m/Q lower with a much higher signal, and these two species would correlate well. L245/246: Schobesberger et al. (2015) provide a detailed list of observed signals in the nucleating system of sulfuric acid and ammonia. From their observations prominent signals for 3 the reported masses (m/z 344 and m/z 362) seem rather unlikely. I would also be surprised if just these two mixed ammonia-sulfuric acid peaks show up in the spectra without any others. Have the authors considered the isotopic distributions of the assigned signals in their analysis? Sulfur has a distinct isotopic pattern; therefore, the assigned formulas in Table S2 for the sulfur containing species could be checked by considering the isotopes. RESPONSE: Isotopes were considered for all peaks; however, these peaks have been removed from this analysis as they would likely not exist in the absence of smaller peaks of similar composition (see more detail below). These reference points have been replaced with reference to SA-DMA clusters. L267/268: As mentioned before, it would be great if more information on HOx and RO2 could be provided. RESPONSE: See above. L295: the plot does not show concentrations but the raw signals L344: J(O1D) is not shown in Figure S1 L347: neither O3 nor HOM are shown in Figure S2 RESPONSE: This has been fixed. L410: in the PSM particles are grown within the condenser RESPONSE: Corrected. L411 and L412: Can the authors at least speculate what compounds cause these signals? If they are from (in)organic compounds (H2O, NH3, H2SO4 and maybe amines) the number of possible combinations should not be too large. RESPONSE: This has been amended. Signal intensity for these peaks was extremely low and over-represented due to the normalisation that had been applied so this section was discarded. These figures had been amended but the text had not. L420 to 430: The possibility of sulfuric acid-amine nucleation should be further discussed. To me it seems very unlikely that only selected SA-DMA clusters show up in the spectra. For nitrate CI-APi-TOF measurements a detailed study of sulfuric acid-dimethylamine clusters has recently been presented (Kürten et al., 2014). That study has also shown that DMA together with sulfuric acid forms new particles very efficiently; therefore, tiny amounts (pptv) should suffice for efficient nucleation and the presence of DMA in clusters is already evidence that DMA is assisting in NPF. I suggest to search for further DMA (or other amine) containing clusters and to check if ambiguity can be ruled out, e.g., that the clusters with DMA and sulfuric acid are not due to some other (organic) compound. This can be done by taking into account the isotopic patterns. In addition, in Table S2 one of the listed clusters is C2H7NHSO4 ‒ (i.e., a C2-amine clustered with the bisulfate ion). This cluster does, however, not exist as the Lewis base (HSO4 ‒ ) does not form a stable cluster with a strong base (C2-amine) unless at least two further acids (H2SO4) are present in the cluster (Ortega et al., 2014; Kürten et al., 2014). RESPONSE: Peaks that were assigned SA-DMA clusters were very small (and often on the shelves of larger peaks). The isotope patterns were considered but these isotopic peaks were even smaller. The assigned SA-DMA clusters may have been misassigned previously as we are also dubious about the presence of peaks with multiple SA, ammonia and water molecules, while smaller SA-NH3 peaks are not present. However, reconsidering the mass spectra has yielded a handful of useful SA-DMA peaks. Some are still lost to shelves of other peaks and some others are not present. Peaks include

C2H7N(H2SO4)2HSO4-, (C2H7N)2(H2SO4)2HSO4-, and (C2H7N)2(H2SO4)3HSO4- and have been added to the peak list, and these correlate well with each other, as well as sulfuric acid monomer and dimer. SA-NH3 peaks are not present. We have added to and edited the manuscript accordingly. Figure 2: Is this MD plot corresponding to a period when NPF is occurring? It would be good to show a second MD plot for another day (same time of day) when no NPF is occurring just to see what signals could make the difference. In addition, there seem to be really prominent peaks (negative MD) at m/z of ∼500 and ∼700. Have the corresponding compounds been identified? Do these signals show a distinct diurnal pattern with higher concentrations during NPF? RESPONSE: The mass defect plot in the manuscript was initially for the nucleation period across the day of 25/06. This has been amended and the figure now shows the daytime period 10:30 – 12:00 on 25/06 to show a nucleation period, and 23/06 in a non-nucleation period. The HOMs + sulfuric acid monomer show the most significant increase between these two periods, most markedly <400 m/Q.

REVIEWER #2 L26: VOC abbreviation in the abstract is not defined RESPONSE: Added. L28-31: sentence is hard to read RESPONSE: Reworded. L 58. Delete 'the' before 'many' RESPONSE: Fixed. L76: Add a ", the" after compound RESPONSE: There is no mention of the word "compound" on L76, and this would not make sense on any other word in this paragraph? L77: Abbreviate BVOC here. Biogenic should compare with anthropogenic VOCs. So correct these statements. RESPONSE: Corrected. L87-88: Statement is not clear. RESPONSE: Reworded. The point of the statement is that the size and oxygen containing functionalities found in HOMs result in low and extremely low vapour pressures. L129: change "organics" to "organic compounds" RESPONSE: Fixed L111: what is APHH here? RESPONSE: Air Pollution and Human Health in a Developing Megacity. This has been added. L179: what is it stand for d'p and Nd'p here? L186: What is J(O'D)?? Please provide some baseline information here. RESPONSE: d'p is the diameter of the particle, Nd'p is the number of particles at diameter d'p. These have been added, J(O1D) is the photolysis rate of ozone, but references to this were erroneous and have been removed. L244-258: This

text is not clear enough to understand the Figure 2. Some background information should be provided how to interpret the data and what is observed followed by what does it mean? RESPONSE: The background information is to be found in the figure caption, but this has also been added to the main body of text. L261: change 'throw light upon' with 'reveal' RESPONSE: Changed. L263: peak in the daytime? But it looks from Figure 3 that mostly peaking in the evening/night time. Why there is no peak on 23-24/06/2017 RESPONSE: On 24/05 this peak is to be found in the afternoon. On the subsequent day there are two peaks (one just before midday, one shortly afterwards). We regarded all of these as daytime peaks. We presume the reviewer means 22/06-23/06? Light intensity was significantly lower on these two days than on other days of measurement, and temperatures were lower also, so both OH. would be lower, and lower temperatures limit the rates of autoxidation. L268: HOM components peaking in the daytime? From Figure 3, it is not clear. What is the basis for this assumption that HOMs are produced by the oxidation of anthropogenic/biogenic components (e.g., alkylbenzenes, monoterpenes, isoprene). Figure 3: C6 - C9 components, and summed C11 - C18 components, assumed to be dominated by alkylbenzenes and other larger components respectively-how this was assumed? RESPONSE: The compounds were grouped by both their molecular formulae, as the HOM products of the oxidation of, for example, xylenes have been studied in flow tubes (forming largely compounds of formulae C8H12Ox) and are therefore known. Aromatics like alkylbenzenes, as well as naphthalene and biphenyl, alongside isoprene and monoterpenes are currently the only molecules known to produce HOM. We also know the abundances of alkylbenzenes, monoterpenes, isoprene and other VOCs from PTR-MS. Earlier in the campaign and not coincidental with our CI-APi-ToF data, GC and 2DGC VOC data were collected and these were also used to take a broad view on the relative abundances of different VOCs with the same mass (ie, how much limonene as compared to alpha-pinene, or how much ethylbenzene vs xylenes, with xylenes having significantly higher HOM yields). This information was used to conclude that alkylbenzenes, monoterpenes and isoprene produce most of the observed HOM, and further to link individual HOM to

their precursor VOCs. C11-18 compounds were assigned as individual oxidation products of single larger VOC rather than the dimers of smaller RO2 molecules due largely to their C:H ratios being small, and indicative of aromatic precursors, and secondarily due to the small fraction of dimers seen of, for example, monoterpene oxidation products. Further to this, although fragmentation upon reaction with OH, or even upon secondary reaction with O3 in the case of molecules with multiple double bonds like limonene occurs and can produce products with lower carbon numbers (ie, C9H14Ox can either be a product of mesitylene oxidation or monoterpene oxidation), the bulk of these signals seemed not to come from fragmentation as what could be assigned as a possible fragment tended to always correlate poorly with both the VOC from which it would have fragmented, and the other oxidation products of that VOC.

L 284: Please add panels 'a, b, and c' in Figure 3 and accordingly refer these in the discussion in the manuscript here. RESPONSE: This has been added. L312: Relative to what?? RESPONSE: The relative ratios of C8H10On where n=5,6,7... This has been reworded for clarity. L320: majority of peaks occurring the daytime? But from Figure 3, it is mostly in evening time. RESPONSE: See above, the actual peaks are specified. L336: Early afternoon peak? By looking at Figure 4, it looks like evening hrs. The scale showed '0:00 HRS' – Is it 24:00 hr? RESPONSE: 00:00 is midnight. This is referring to the HOM peak, which is at 16:00 on 24/06 and 15:00 on 25/06. Early afternoon is probably an incorrect choice of wording here, this has been amended. L337-338: From where, it is inferred this (i.e., similar behaviour of C3-benzenes and their oxidation products as C2-benzenes and their HOMs)? RESPONSE: We have reworded this to clarify. L344: I could not find J(O'D) in Figure S1! RESPONSE: J(O1D) data was removed in an earlier version of this paper, this reference was erroneous. L346-347: Figure S2 does not provide this information, please double check and maintain consistency between text and supplementary information. RESPONSE: This has been fixed. L350: This is inferred from which figure, please mention RESPONSE: Figure 3b, this has been added. L369: text is unclear-'what is unit mass data' RESPONSE: Unit mass refers to mass spectral peak area data integrated over the whole of one unit mass,

producing less complicated low resolution data. L371-372: This normalization part is not clear enough to follow the figure 5. Please elaborate. RESPONSE: Reworded for clarity. L403-404: sentence is not clear. In Figure 3, point the two peak of HOMs on 25/06/2017 to understand the text described here. RESPONSE: This has been streamlined and better explained in the text. L405: I am not able to see the two peaks in C2-benzenes in Figure 4. Please encircle or write clearly to maintain consistency with figure. RESPONSE: This was an erroneous reference and has been amended. L407-9: These sentences are not clear. Please consider rephrasing these sentences RESPONSE: Done. L412: From Figure 5, PSM cluster peaked at 10:00 and 13:00 h have m/Q between 200-550 (as also stated on L409). But the specified m/Q here is beyond the scale shown in Figure 5. Is it correct or I am missing something. RESPONSE: This has been amended. Signal intensity for these peaks was extremely low and over-represented due to the normalisation that had been applied so this section was discarded. These figures had been amended but the text had not. L412: Add 'because of' after 'presumably' RESPONSE: See above. L416- 418: Please refer to the figure. RESPONSE: These figures are referred to above. L425: Define 'SA-DMA' here. RESPONSE: Added. L443: From the Figure 3, HOM peaked in the evening hours on 24/06/2017 compared to 25/06/2017, where HOM peaked at the early afternoon. So 'daytime peak of HOMs' need to be rephrased RESPONSE: This has been rephrased.

---

## Referee Report (RR1)

The authors did a good job in considering the comments from both referees. I therefore recommend that the paper should be published soon in ACP after considering the following minor comments:

1) I think it is confusing to report normalized signals, which are multiplied by an arbitrary factor of 10^5. It would be better to omit such a scaling factor.

2) L67: "the production"

3) L300: Please delete the plus sign behind "C5".

4) L372: It would be good to discuss briefly why there is little correlation between VOCs and HOMs during the first campaign days.

5) L449: Isn't it rather a "morning" peak instead of an "afternoon" peak?

6) L478: "sulfuric acid molecules" instead of "sulfuric acid ions".

7) Figure 2: Panel B shows two very prominent signals. It would be good to mention what species cause these signals.

8) Figure 3: The figure does not show the "concentrations" of HOM but the sum of the normalized signals.

---

## Author Response (AR2)

**Journal: ACP**
**MS No.: acp-2019-156**
**MS Type: Research article**
**Title: Observations of Highly Oxidised Molecules and Particle Nucleation in the Atmosphere of Beijing**
**Author(s): James Brean et al.**

**RESPONSE TO CO-EDITOR**

Lines 181 and 351. It would be helpful for the readers to better understand the importance of this study if authors could add a few references that have reported the presence of oxalic and malonic acids in ambient aerosols from Beijing and other cities.

A brief discussion of the sources of dicarboxylic acid have been added with a few relevant publications. The section reads as follows

*"Measurements of particle phase dicarboxylic acids in cities typically show greater concentrations of of oxalic acid than malonic (Ho et al., 2016), and these acids are primarily produced in the aqueous phase (Bikkina et al., 2014). Primary sources of dicarboxylic acid include fossil fuel combustion (Kawamura and Kaplan, 1987) and biomass burning (Narukawa et al., 1999), which are both plentiful in urban Beijing."*

**Other edits made:**
Mention of the $10^5$ correction factor applied to CI-APi-ToF signals was removed from methods section (as it was removed from all figures)

---

## Author Response (AR3)

**Journal: ACP**
**MS No.: acp-2019-156**
**MS Type: Research article**
**Title: Observations of Highly Oxidised Molecules and Particle Nucleation in the Atmosphere of Beijing**
**Author(s): James Brean et al.**

**RESPONSE TO CO-EDITOR**

**Minor Revision**

> **Co-Editor Decision: Publish subject to technical corrections** (29 Oct 2019) by Kimitaka Kawamura
> Comments to the Author:
> Bikkina et al. (2014) is described in line 335, but it is missing in the Reference section. Please make an addition of the reference.
>
> Non-public comments to the Author:
> Some chemical forms in supplement tables are not formatted to subscript.

Bikkina et al. (2014) is described in line 335, but it is missing in the Reference section. Please make an addition of the reference
**RESPONSE:** This reference has been added, for reference:

Bikkina, S., K. Kawamura, Y. Miyazaki, and P. Fu (2014), High abundances of oxalic, azelaic, and glyoxylic acids and methylglyoxal in the open ocean with high biological activity: Implication for secondary OA formation from isoprene, Geophys. Res. Lett.,41, 3649–3657, doi:10.1002/2014GL059913.

Some chemical forms in supplement tables are not formatted to subscript
**RESPONSE:** This has been fixed

[revised manuscript text omitted]